# Psychometric properties of novel instrument for evaluating ambient air pollution health literacy in adults

I-Chen Chen[1,2], Chung-Yi Li[2,3,4], Chien-Yeh Lu[5], Yi-Chin Huang[2], Pei-Chen Lee[2], Ming-Yeng Lin[6], Yu-Chen Wang[7], Long-Sheng Chen[8], Chia-Lun Kuo[2,9☯*], Wen-Hsuan Hou[2,10,11☯*]

1 Department of Occupational Therapy, College of Nursing and Health Sciences, Da-Yeh University, Changhua, Taiwan, 2 Department of Public Health, College of Medicine, National Cheng Kung University, Tainan, Taiwan, 3 Department of Public Health, College of Public Health, China Medical University, Taichung, Taiwan, 4 Department of Healthcare Administration, College of Medical and Health Science, Asia University, Taichung City, Taiwan, 5 School of Gerontology Health Management & Master Program in Long-Term Care, College of Nursing, Taipei Medical University, Taipei, Taiwan, 6 Department of Environmental and Occupational Health, College of Medicine, National Cheng Kung University, Tainan, Taiwan, 7 Department of Law, College of Social Science, National Cheng Kung University, Tainan, Taiwan, 8 Surveillance, Research and Health Education Division, Health Promotion Administration, Ministry of Health and Welfare, Taipei, Taiwan, 9 Department of Psychiatry, Tsaotun Psychiatric Center, Ministry of Health and Welfare, Nantou, Taiwan, 10 Department of Geriatrics and Gerontology, National Cheng Kung University Hospital, College of Medicine, National Cheng Kung University, Tainan, Taiwan, 11 International Ph.D. Program in Gerontology and Long-Term Care, College of Nursing, Taipei Medical University, Taipei, Taiwan

☯ These authors contributed equally to this work.
* houwh@gs.ncku.tw (WHH); ichbinshirley@gmail.com (CLK)

**Data Availability Statement:** All relevant data are within the manuscript and its Supporting Information files.

## Abstract

We aimed to develop and validate a comprehensive ambient air pollution health literacy instrument. We developed items covering 12 constructs, four information competencies within three health domains. In this population-based telephone interview study, probability proportional to size sampling and random digit dialing were used to determine participants. We conducted confirmatory factor analysis to analyze model fits and used content validity indices and Cronbach's alpha to measure content validity and internal consistency reliability. Twenty-four items were generated, and a total of 1,297 participants were recruited. A theoretically conceived 12-factor model was supported (root mean square error of approximation [RMSEA] = 0.068, comparative fit index [CFI] = 0.039, standardized root mean square residual [SRMR] = 0.934, normed fit index [NFI] = 0.914, Tucker–Lewis index [TLI] = 0.902). Content validity indices for relevance, importance, and unambiguity were 0.97, 0.99, and 0.94, respectively. Internal consistency reliability assessed by Cronbach's alpha was 0.93. The ambient air pollution health literacy instrument is valid and reliable and can be used in community residents. The novel instrument can guide the stakeholders and the authority to tailor and implement effective and appropriate interventions and actions, empowering the public to manage hazardous exposure and improving AAPHL of the public.

**Funding:** The authors are grateful for grants from Health Promotion Administration, Ministry of Health and Welfare (109-0331-02-18-04). The funder has no role in conducting and submitting this work.

**Competing interests:** The authors have declared that no competing interests exist.

## Introduction

The World Health Organization claims the premature deaths of 4.2 million people worldwide per year are attributable to ambient air pollution (AAP) [1]. A growing body of evidence has demonstrated AAP has adverse effects on cardiovascular and respiratory systems [2], central nervous system [3], cancers [2], and children and newborn development [4]. Similarly, in Taiwan, hazardous impacts of AAP on health are crucial and urgent issues to be addressed. According to the world's most polluted countries of 2020 World Air Quality Report, Taiwan ranks no. 62 with estimated annual average fine particulate matter ($PM_{2.5}$) concentration of 15.0 μg/m$^3$ among 106 countries/regions [5]. Prior studies have found the associations between increased burden of disease and AAP, and compromised systems involve in central nervous system [6], cardiovascular, and respiratory systems [6, 7]. Significant evidence shows AAP causes numerous and diverse diseases; thus, how to reduce exposure to threatening environment is critical to maintain good health.

Given that AAP poses substantial threats to health, highlighting the importance of preventing exposure to harmful air pollutant is crucial for the public in Taiwan. Finn and O'Fallon (2017) suggest environmental health literacy (EHL) as "a progressive nature of EHL beginning with an individual's understanding of the connection between environmental exposure and human health and proceeding to the ability to create effective public health messages and/or act on this understanding." EHL focuses on improving the understanding of environmental threats and on preventing adverse health outcomes, which are environmentally induced [8, 9]. Therefore, raising EHL is involved in empowering the public to search for, comprehend, and evaluate environmental health information, and make efforts to reduce unhealthy environmental exposure.

Validated measures are essential to improve EHL for the public by identifying competencies and deficiencies and guiding the authority to develop effective policies for improving EHL of the public. Although numerous general health literacy surveys exist, only a few surveys are related to ambient air pollution health literacy (AAPHL). Lichtveld et al. (2019) develop scales to assess knowledge (fact and information), attitudes (thinking and feeling), and behaviors (how to response to) for air, food, water, and general EHL of graduate and undergraduate students who enroll in three different courses from the same public health program [10]. Similarly, Odonkor and Mahami (2020) develop a survey to evaluate knowledge and attitudes for air pollution and general EHL of residents who live in one district [11]. The two surveys containing cognitive, attitudinal, and action dimensions are based on behavior change theories/models [12, 13]. However, EHL recently focuses on the ability to seek out, understand, and judge information to make decisions and actions for preserving good health and controlling the environment [9, 14]. Moreover, the participants in the two studies restricted to specific populations and the findings are vulnerable to be generalized.

Individuals with good EHL are more likely to protect themselves from hazardous exposure and to maintain good health and manage surrounding environments. To improve AAPHL of the public, a validated AAPHL instrument can identify its deficiencies, navigate the effective policies to be developed. However, currently, no specific tool that is designed for assessment of AAPHL exists in Taiwan or in the world. Existing air-pollution-related EHL surveys are based on behavior change theories/ models, rather than concept/model of EHL, and generalization of the findings is limited due to selection of subjects. Consequently, the present study aims to develop and validate a Chinese version instrument based on the integrated conceptual model of health literacy to evaluate individuals' AAPHL through a population-based study design.

## Materials and methods

### Instrument development

The process of the Chinese version of AAPHL instrument development and validation was presented in Fig 1.

### Literature review and domain identification

The Chinese version of AAPHL instrument development started with an integrated conceptual model of health literacy [15] and was based on European Health Literacy Survey Questionnaire [16]. The integrated conceptual model of health literacy comprises 12 dimensions, a combination of four health information processing competencies within three domains of health continuum across life course. The four competencies are to access, understand, appraise, and apply information, and the three domains of health continuum refer to healthcare, disease prevention, and health promotion [15].

### Item generation and instrument formation

Each estimated construct (dimension) was defined by two measures (items) [17] and a total of 24 items were generated for the Chinese version of APPHL instrument. Interviewees

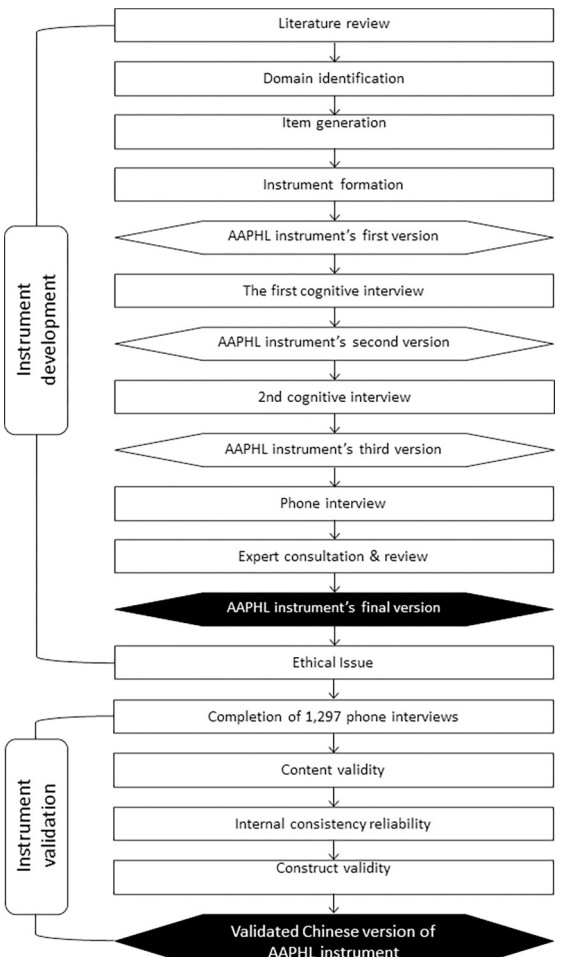

**Fig 1. Process of the Chinese version of AAPHL instrument development and validation.**

responded to 24 items using a four-point Likert-type scale, 1 = "*very difficult*," 2 = "*fairly difficult*," 3 = "*fairly easy*," and 4 = "*very easy*." If response of an interviewee was "*I do not know*" or "*I have no experience*," then item was coded as a missing value [18] during data analysis. A composite score for each dimension was calculated as a mean of two corresponding items, and a total score for the Chinese version of APPHL was calculated as a mean of 12 dimensions. Higher scores indicated better competency.

Given the potential effects of personal, situational, as well as societal and environmental determinants on health literacy [15], we included 7 items related with demographic characteristics as personal determinants: sex, age, level of education and occupation. Situational determinants were living arrangement and marital status. Societal and environmental determinants referred to habitat.

Six experts were recruited to check the new instrument. Two experts are professionals in epidemiology, two experts are professionals in environmental health studies, and two are lay experts. They reviewed instructions for instrument, wording of items, structure of items, order of items, and response alternatives. Given the protocol of a phone interview and avoidance of any confusing and chaotic situation, the order of 24 items was fixed, rather than random order. Then, the first Chinese version of AAPHL instrument was completed. The period of time between conception and completion of the Chinese version of AAPHL was from June 1st, 2020 and July 24th, 2020.

## Cognitive interviews

Cognitive interviews were conducted to learn how the interviewees understood items and how they organized their responses; they were also used to investigate whether the instrument appeared to test what it was supposed to and to ask feedback from interviewees for instrument revision [19]. We invited 17 interviewees above 20 years of age for the first face-to-face cognitive interviews. According to the feedback from the interviewees, a detailed instruction about the concepts of four information competencies within three health contexts was added; information regarding specific laws and regulations was provided if the description of item included the Air Pollution Control Act. The second versions of AAPHL were completed. Then, we invited 16 interviewees who were above 20 years of age and not involved in the first interview for the second face-to-face cognitive interviews. After the second cognitive interviews, examples of laws and regulations were also provided to make interviewees understand the descriptions easily; we revised descriptions with straightforward sentences instead of sentences with double negatives. The third versions of AAPHL were completed. Next, the phone interview organization invited 10 interviewees for phone interview using the third version of the AAPHL instrument to ensure that the descriptions were appropriate.

## Expert consultation and review

Expert consultation was organized to review the items and to evaluate the content validity. Three experts were invited as advisory panel; two experts are professionals in epidemiology, and one expert is a professional in environmental health studies. Three experts assessed whether the instrument collected information needed to measure the intended concepts, and whether the instrument met content, cognitive, and usability standards [19]. They used a four-point Likert-type scale to evaluate whether the items were relevant, important, and unambiguous; the higher points indicated the better fits for relevancy, importance, and unambiguity. After expert consultation and review, we reviewed each item, eliminated redundant words, and modified ambiguous wording. The final Chinese version of AAPHL instrument was completed.

## Ethical issue

The ethical approval of the final version of AAPHL instrument was obtained from Institutional Review Board of National Cheng Kung University Governance Framework for Human Research Ethics (#109–385).

## Instrument validation

**Research design.**   In this study, a population-based, cross-sectional, random-digit dialing telephone survey was conducted, we used a probability proportional to size sampling, which was proportional to habitat strata, to build a nationally representative sample for sound external validity. In Taiwan, the number of mobile-only users is highly increasing; these residents cannot be reached by landline and that would lead to large under-coverage rates when phone samples only from landline frame. Thus, we used a dual frame approach to include samples from landline and mobile frames. For landline frame, the area codes of 22 cities and counties are specific for each city and county. For mobile frame, given that mobile numbers are not city- or county-specific, samples of mobile were selected across the entire country. According to distribution of geographic areas, whole population was grouped into five strata, northern (43.66%, six cities and counties), central (24.34%, five cities and counties), southern (26.66%, five cities and counties), eastern (4.19%, three counties) areas, and outlying islands (1.08%, three counties). Participants aged between 20 and 70 years were included. Based on age distribution, whole population is grouped into three strata, ages 20–39 years (38.2%), ages 40–54 years (32.1%), and ages 55–70 years (29.6%).

We used computer-assisted telephone interviewing (CATI) technique. Five trained interviewers conducted CATI from 5 PM to 10 PM during period between September 11[th], 2020 and October, 22[nd], 2020. The interviews lasted approximately 15–20 minutes. After ensuring interviewee's eligibility, the participants were invited to our study and verbal consents were obtained. Each landline/mobile phone call included only one participant.

**Statistical analysis.**   Descriptive statistics were used to describe the characteristics of all participants. Content validity index (CVI) is the most used for content validity in instrument development. The value of CVI ranges from 0 to 1; if CVI is greater than 0.79, then the item is appropriate [20]. We calculated Cronbach's alpha to examine internal consistency reliability, and the acceptable value of Cronbach's alpha is 0.7 or greater [21].

Confirmatory factor analysis was used to examine model fit of three models: (1) four-factor model (four information competencies: accessing, understanding, appraising, and applying information); (2) three-factor model (three domains of health continuum: healthcare, disease prevention, and health promotion); (3) 12-factor model (four information competencies within three domains of health continuum).

We assessed the overall fit of model and fit of internal structure of model. For global measures of fit, we used absolute fit indices (root mean square error of approximation [RMSEA] and standardized root mean square residual [SRMR]) and incremental fit indices (comparative fit index [CFI], normed fit index [NFI], and Tucker–Lewis index [TLI]). The acceptable values of fit indices are 0.08 or less for RMSEA [22], 0.08 or less for SRMR [23], 0.9 or greater for CFI [24], 0.9 or greater for NFI [25], and 0.9 or greater for TLI [26]. For the fit of internal structure of model, we measured composite reliability (CR) and average variance extracted (AVE) for each construct; the acceptable values of CR and AVE are 0.6 or greater [27] and 0.5 or greater [27, 28], respectively.

## Results

### Response rates of phone interviews

Numbers of successful landline and mobile phone interviews for the northern, central, southern, eastern, and outlying islands strata were 639 (49.11%), 285 (21.90%), 303 (23.29%), 60

(4.61%), and 14 (1.08%), respectively. Numbers of participants who successfully completed phone interviews for ages 20–39 years, ages 40–54 years, and ages 55–70 years strata were 437 (33.59%), 414 (31.82%), and 450 (34.59%), respectively.

CATI procedure reached 4,084 eligible adults via landline, and 1,072 adults successfully completed the interviews, with a response rate of 26.25%; it researched 835 eligible adults via mobile and 225 adults successfully completed the interviews, with a response rate of 26.95%. Both landline and mobile phone interviews had similar response rates. Reasons for unsuccessful phone interviews were mainly because interviewees thought the interview took more time than they expected and decided not to continue (19.9%), or interviewees declined to be interviewed before interview began (80.1%).

### Demographic characteristics of participants

A total of 1,297 eligible participants were recruited in this study and analyzed in the following analyses. The mean age of participants was 46.89 (standard deviation [SD] = 13.89) years, 55% were female, 61% completed college or higher degrees, 65% were married, and 89% lived with someone else. The demographics of all participants were presented in Table 1.

### Psychometric properties and model fits

For content validity, the mean scores of relevance, importance, and unambiguity were 3.90, 3.97, and 3.82, respectively, judged by three experts. The values of CVI for relevance, importance, and unambiguity were 0.97, 0.99, and 0.94, respectively. The internal consistency reliability calculated with Cronbach's alpha was 0.93.

The overall model fit and the fit of internal structure of 12-factor model were fairly satisfactory. The 12-factor model was fitted to the 24 items: RMSEA = 0.068, CFI = 0.039, SRMR = 0.934, NFI = 0.914, and TLI = 0.902 (Table 2). Meanwhile, three- and four-factor models were slightly below the acceptable critical values for model fits (Table 2).

Sex-stratified and age subgroup analyses of model fits were presented in S1 and S2 Tables, respectively. Subgroups of male and female demonstrated similar trends in the three different models and the model fits of 12-factor model were acceptable. Subgroup of aged younger than 65 showed that model fits of 12-factor model was fair. On the other hand, subgroup of aged 65 or over did not demonstrate acceptable model fits in any models.

The standardized factor loadings of items ranged from 0.61 to 0.88. The values of CR for each construct ranged from 0.56 to 0.84, and values of AVE for each construct ranged from 0.39 to 0.71 (Table 3).

### Scores of AAPHL instrument

A mean of total score was 2.87 (SD = 0.57). Among the 12 dimensions, the highest score was 3.21 of applying in disease prevention, and the lowest score was 2.60 of appraising in disease prevention (Table 4).

\Sex-stratified and age subgroup analyses of scores were presented in S3 and S4 Tables, respectively. Subgroups of male and female demonstrated similar performance across the 12 dimensions and the dimension with highest score was applying in disease prevention. Subgroup of aged 65 or over showed higher total score than subgroup of aged younger than 65. The dimension with highest score in the subgroup of aged 65 or over was applying in health promotion while subgroup of aged younger than 65 had the highest score in applying in disease prevention.

**Table 1. Demographic characteristics of participants (N = 1,297).**

| Variable | n | (%) |
|---|---|---|
| Sex | | |
| Male | 576 | (44.4) |
| Female | 721 | (55.6) |
| Age | | |
| 20–29 | 191 | (14.7) |
| 30–39 | 245 | (18.9) |
| 40–49 | 230 | (17.7) |
| 50–59 | 356 | (27.5) |
| 60–70 | 275 | (21.2) |
| Education level | | |
| Middle school or less | 125 | (9.6) |
| High school | 373 | (28.8) |
| College/Undergraduate | 697 | (53.7) |
| Graduate | 102 | (7.9) |
| Occupation | | |
| Finance, banking, insurance, real estate | 82 | (6.4) |
| Industrial production, manufacture commercial, shop, buy and sell | 264 | (20.6) |
| Information technology, automation | 71 | (5.5) |
| Construction, fittings, housing | 65 | (5.1) |
| Accommodation, food preparation and servicing | 89 | (6.9) |
| Transport, logistics, port, airport, telecommunication | 146 | (11.4) |
| Culture, design, sports, tourism, leisure, education, research, training Healthcare, care, welfare, social work | 39 | (3.0) |
| Government administration, policy adviser, military affairs, nonprofit organization | 51 | (4.0) |
| Others (e.g., retiree, student, housekeeper) | 423 | (33.0) |
| Living arrangement | | |
| Living alone | 147 | (11.3) |
| Living with someone else[a] | 1150 | (88.7) |
| Children under age 12 | 269 | (20.9) |
| Person who is a student above 12 years of age | 509 | (39.5) |
| Person above 65 years of age | 510 | (39.6) |
| Marital status | | |
| Single | 361 | (27.8) |
| Married | 843 | (65.0) |
| Divorced/Separated | 73 | (5.6) |
| Widowed | 20 | (1.5) |
| Habitat in Taiwan | | |
| North | 636 | (49.0) |
| Central | 285 | (22.0) |
| South | 303 | (23.4) |
| East | 60 | (4.6) |
| Outlying islands | 14 | (1.1) |

*Note.* [a] Living with someone else is a multiple choice. That is, an interviewee may live with a child under age 12 and a person above 65 years of age. Thus, the total number of children under age 12, person who is a student above 12 years of age, and person above 65 years of age (n = 1,288), is greater than the number of living with someone else (n = 1,150).

**Table 2. Results of statistical fit indices for the confirmatory factor analysis.**

| Fit index | 4-factor model | 3-factor model | 12-factor model | Critical value |
|---|---|---|---|---|
| Absolute fit indices | | | | |
| RMSEA | 0.088 | 0.089 | 0.068 | $\leq 0.08$ |
| SRMR | 0.062 | 0.058 | 0.039 | $\leq 0.08$ |
| Incremental fit indices | | | | |
| CFI | 0.847 | 0.841 | 0.934 | $\geq 0.90$ |
| NFI | 0.822 | 0.817 | 0.914 | $\geq 0.90$ |
| TLI | 0.828 | 0.824 | 0.902 | $\geq 0.90$ |

*Note.* CFI, comparative fit index; NFI, normed fit index; RMSEA, root mean square error of approximation; SRMR, standardized root mean square residual; TLI, Tucker–Lewis index.

## Discussion

This was the first study to develop and validate a novel measure of AAPHL based on model of health literacy with sound psychometric properties from a nationally representative sample. A total of 24 items representing the constructs were carefully developed and the items designed to measure the 12 constructs had fair psychometric properties. This instrument is ready to be employed for research evaluating individuals' competencies to access, understand, appraise, and apply environmental health information within healthcare, disease prevention, and health promotion contexts based on the integrated conceptual model of health literacy [15].

### Psychometric properties and model fits

The results suggest the 24 items are appropriate and the instrument has sound internal consistency. We calculated CVI for content validity and the values of CVI were greater than 0.79, the acceptable values of CVI [20]. Since none of item was rated as irrelevant, unessential or unclear, all items were included in this instrument. We calculated the coefficient of Cronbach's alpha for internal consistency and the coefficient of Cronbach's alpha was 0.93, greater than the ideal value of 0.7 [21]. Bland and Altman suggest that critical values of Cronbach's alpha may vary, which depends on different situations, and that value between 0.90 and 0.95 is desirable for clinical application [29]. However, Streiner suggest that values of Cronbach's alpha greater than 0.90 may be too high and implies the potential redundancy among items [30].

The AAPHL instrument with 24 items has acceptable model fit and fair psychometric properties; the results indicate the items assessing the 12 constructs are relevant to AAPHL, and none of the items should be eliminated. That is, the 12-factor structure of developed AAPHL instrument confirms the hypothesis of the integrated conceptual model of health literacy comprising four information competencies within three domains of health continuum. In the 12-factor model, all standardized factor loadings ranged between 0.61 and 0.88, greater than the ideal values of 0.7 [28]. Since none of factor loading was below 0.5, all items were included.

The results indicate that the constructs are well measured/ explained by the developed items and reliable to measure AAPHL. We calculated AVE for each construct, and the values of AVE for most constructs were greater than the acceptable value of 0.5 [27, 28]. However, the values of AVE pertaining to understanding, appraising, and applying in health promotion are slightly less than 0.5. We calculated CR for each construct, and the values of CR for all constructs were greater than the acceptable value of 0.6 [27], except for the appraising of healthcare.

**Table 3. Psychometric properties of the final APPHL instrument.**

| # | Item | FL | AVE | CR |
|---|------|----|----|----|
| | Accessing in Healthcare | | 0.71 | 0.83 |
| 1 | Is it easy for you to access information when you want to know the influence of air pollution on health? | 0.80 | | |
| 2 | Is it easy for you to access information when you want to know the status of air quality? | 0.88 | | |
| | Understanding in Healthcare | | 0.52 | 0.68 |
| 3 | Is it easy for you to understand the air quality index provided on the website of EPA, the weather reports of Central Weather Bureau or the information released by the government? | 0.71 | | |
| 4 | Is it easy for you to understand the influences of poor air quality on health? | 0.73 | | |
| | Appraising in Healthcare | | 0.39 | 0.56 |
| 5 | Is it easy for you to evaluate the air quality around your living environments (including the community and neighborhood you live in)? | 0.64 | | |
| 6 | Is it easy for you to judge whether the suggestions for improving air quality provided by the social media are incorrect? | 0.61 | | |
| | Applying in Healthcare | | 0.50 | 0.67 |
| 7 | Is it easy for you to engage in the proper outdoor activities based on the air quality index provided by EPA? | 0.77 | | |
| 8 | Is it easy for you to adopt the suggestions for air pollution improvement put forward by the government? | 0.64 | | |
| | Accessing in Disease Prevention | | 0.57 | 0.72 |
| 9 | Is it easy for you to access information about reducing the malign effects of air pollution on health? | 0.80 | | |
| 10 | Is it easy for you to learn what the government has done to reduce air pollution? | 0.70 | | |
| | Understanding in Disease Prevention | | 0.59 | 0.75 |
| 11 | Do you know which air pollutant is harmful to human health? | 0.79 | | |
| 12 | Can you understand why the government monitors the level of outdoor air pollution? | 0.76 | | |
| | Appraising in Disease Prevention | | 0.73 | 0.84 |
| 13 | Can you judge whether the information regarding the disease caused by air pollution provided by social media is correct? | 0.84 | | |
| 14 | Can you judge whether the approaches to decreasing the malign effects of air pollution on health provided by social media are credible? | 0.86 | | |
| | Applying in Disease Prevention | | 0.50 | 0.66 |
| 15 | Do you know how to decrease the possibility of diseases caused by air pollution by yourself? (e.g., When do you need to decrease outdoor activities? Where are the proper places for outdoor activities?) | 0.79 | | |
| 16 | Do you have your vehicles inspected regularly to reduce air pollution and potential diseases caused by air pollution within the time limit set by the government? | 0.62 | | |
| | Accessing in Health Promotion | | 0.52 | 0.68 |
| 17 | Can you access the government laws and regulations regarding air pollution control? | 0.74 | | |
| 18 | Can you access information about how to improve the air quality of your surrounding? | 0.70 | | |
| | Understanding in Health Promotion | | 0.48 | 0.65 |
| 19 | Do you know that the government provides subsidies for the purchase of electric vehicles to reduce air pollution? | 0.64 | | |
| 20 | Can you understand the content of the Air Pollution Control Act in Taiwan? | 0.74 | | |
| | Appraising in Health Promotion | | 0.48 | 0.65 |
| 21 | Can you judge whether the approaches to reducing the air pollution for health promotion provided by the television or online media are correct? | 0.72 | | |
| 22 | Can you judge whether you are eligible to apply for air policy reduction subsidies proposed by the government? | 0.66 | | |
| | Applying in Health Promotion | | 0.46 | 0.63 |
| 23 | Do you usually walk on smoke-free paths, act in smoke-free parks or engage in the activities which take place in smoke-free spaces to maintain health? | 0.67 | | |

(*Continued*)

**Table 3.** (Continued)

| # | Item | FL | AVE | CR |
|---|------|-----|-----|-----|
| 24 | Can you act for your health (e.g., wearing a mask) and for reducing air pollution (e.g., using public transportation system instead of driving your own vehicle)? | 0.70 | | |

*Note*. AVE, average variance extracted (> 0.5 is acceptable); CR, composite reliability (> 0.6 is acceptable); FL, factor loading (> 0.5 is acceptable).

## Scores of AAPHL instrument and implications

Improving the AAPHL of the public is beneficial for individual health maintenance and public health. The validated AAPHL instrument identifies deficiencies of the public that may contribute to poor health management and be needed to be intervened. In this study, among the 12 dimensions, the dimension with the lowest score is "appraising in disease prevention," indicating the appraisal competency of the public is needed to be improved urgently. Our finding is consistent with the previous studies in Japan and Europe, which show appraising health information is the most difficult competency. Appraising information is considered as a complex health literacy competency. Although the valuable and reliable websites exist, these are not always accessible, understandable, or usable by the public with low health literacy. Then, it may become difficult for the public to judge information regarding health [18, 31, 32]. On the other hand, the dimension with the highest score is "applying in disease prevention." Yet, our finding conflicts with the prior studies, which demonstrate the best competency is understanding health information. Information-processing of understanding is considered as a basic health literacy competency. Understanding health information is highly associated with educational level and socio-economic characteristics [18, 31, 32]. However, in this study, the item of dimension of "applying in disease prevention" involves Air Pollution Control Act [33], which requests drivers to conduct vehicle inspections regularly otherwise to be fined. The strict law and concrete penalty make the public obey and perform, and it may explain the reason for the competency of applying health information with the highest score in this study and the inconsistency between our findings and previous studies. The AAPHL instrument can provide stakeholders with the profiles of capacities and the needs of the public, which provides

**Table 4. Descriptive statistics of the 12 dimensions for the AAPHL instrument.**

| Dimension | All participants (*N* = 1,297) | |
|---|---|---|
| | Mean | (*SD*) |
| Total | 2.87 | (0.57) |
| Accessing in healthcare | 3.02 | (0.81) |
| Understanding in healthcare | 3.11 | (0.69) |
| Appraising in healthcare | 2.70 | (0.74) |
| Applying in healthcare | 2.82 | (0.77) |
| Accessing in disease prevention | 2.62 | (0.81) |
| Understanding in disease prevention | 2.94 | (0.76) |
| Appraising in disease prevention | 2.60 | (0.81) |
| Applying in disease prevention | 3.21 | (0.67) |
| Accessing in health promotion | 2.70 | (0.81) |
| Understanding in health promotion | 2.85 | (0.76) |
| Appraising in health promotion | 2.87 | (0.78) |
| Applying in health promotion | 3.12 | (0.75) |

information about what interventions might be needed to ameliorate health outcomes and new insight into how to improve AAPHL of the public.

Raising AAP awareness, increasing AAPHL, and making behavior change are important for the public to be protected from hazardous exposure and to promote health. Communication strategies about AAP are critical to empower the public to act for reducing exposure to environmental threats. Yet, existing communication systems generally have challenges due to the lack of accurate information sources, sound information quality, and timely information reach [34]. Therefore, the authority/government should introduce to correct information resources and disseminate accurate and concise AAP knowledge using plain language with diagrams through systematic approaches and channels [35]. Recently, infographics and memes via social media have become more popular than ever in Taiwan, and both can serve as potential approaches to providing accurate AAP information to the public. Citizen science approach is proposed to enhance AAPHL and to increase community engagement. It contains an interactive air pollution questionnaire, which provides participants with knowledge about AAP and AAP exposure estimation and air quality monitoring with low-cost sensors that participants implement. The approach presents a perception and an estimated AAP exposure of participants and enhances awareness and knowledge of AAP of participants [36]. In Taiwan, EdiGreen Airbox with Location Aware Sensing System has been developed, and more than 4,000 EdiGreen Airboxes are distributed all over the country to monitor and record air quality data (https://airbox.edimaxcloud.com/). A citizen scientist workshop, integrating this AAPHL instrument and community air monitoring with EdiGreen Airbox, can be a feasible and proper intervention to improve AAPHL for the public in Taiwan.

## Strengths and limitations

There are some strengths in this study. First, we applied the health literacy model to develop and validate a novel AAPHL instrument. Second, we tested the psychometric properties of the novel instrument using nationally representative sample. However, there are certain limitations. First, we intended to develop an AAPHL instrument by conducting population-based phone interviews with probability proportional to size sampling. According to data of the Department of Household Registration in 2021, 50.47%, 43.76%, and 38.59% of the population are female, married, and had college or higher educational level, respectively [37]. However, 55.59%, 65.00%, and 61.54% of the participants were female, married, and had college or higher educational level, respectively, which might be due to selective response. The unsatisfactory response rates (approximate 26%) also implied that our findings cannot well represent the responses of nonrespondents. Hence, the participants cannot be a perfect nationally representative sample, which compromises the generalizability of the findings. However, our results show the instrument has sound psychometric properties. Second, health literacy depends on cognitive and communicative/social skills [38], but we did not conduct cognitive or communicative screening tests to ensure the cognitive and communicative function of participants. However, the participants were able to respond to items and to successfully complete phone interviews, which indicated that they possessed basic cognitive and communicative abilities. Additional response categories (i.e., "*I do not know*" or "*I have no experience*") were recorded and coded as missing values during data analysis to preserve participants' comprehension of survey items. Nevertheless, since the AAPHL instrument with sound psychometric properties was validated, cognitive and/or communicative abilities of participants were not issues. Lastly, while EHL varies with cultural contexts and living environments, information regarding where participants resided in, either rural or urban environment, is unavailable., Accordingly, individualized questionnaire with considerations of differences in culture and habitat would be required if apply the Chinese version of AAPHL instrument in other countries.

## Conclusions

This study presents the first validated instrument with 24 items to evaluate individuals' competencies to access, understand, appraise, and apply ambient air pollution related health information within the healthcare, disease prevention, and health promotion contexts. This AAPHL instrument contributes an important and novel instrument to environmental health research and practices. Moreover, the AAPHL instrument can guide the stakeholders and the authority to tailor and implement effective and appropriate interventions and actions, which empower the public to manage hazardous exposure and improve the AAPHL of the public.

## Supporting information

**S1 Table. Results of statistical fit indices for the confirmatory factor analysis (CFA) for sex-stratified subgroup.**
(DOCX)

**S2 Table. Results of statistical fit indices for the CFA for age subgroup.**
(DOCX)

**S3 Table. Descriptive statistics of the 12 dimensions for the AAPHL instrument for sex-stratified subgroup.**
(DOCX)

**S4 Table. Descriptive statistics of the 12 dimensions for the AAPHL instrument for age subgroup.**
(DOCX)

**S1 File.**
(PDF)

## Acknowledgments

We thank all the participants who were involved in this study. Furthermore, we thank Tsaotun Psychiatric Center, Ministry of Health and Welfare of Taiwan for supporting this study. The guarantor is WH Hou who takes full responsibility for the work as a whole, including the study design, access to data, and the decision to submit and publish the manuscript.

## Author Contributions

**Conceptualization:** I-Chen Chen, Chung-Yi Li, Chien-Yeh Lu, Yi-Chin Huang, Pei-Chen Lee, Ming-Yeng Lin, Yu-Chen Wang, Long-Sheng Chen, Chia-Lun Kuo, Wen-Hsuan Hou.

**Data curation:** I-Chen Chen, Chien-Yeh Lu, Yi-Chin Huang.

**Formal analysis:** I-Chen Chen, Chung-Yi Li, Long-Sheng Chen, Wen-Hsuan Hou.

**Funding acquisition:** Chung-Yi Li, Chia-Lun Kuo.

**Methodology:** I-Chen Chen, Chung-Yi Li, Chien-Yeh Lu, Yi-Chin Huang, Pei-Chen Lee, Ming-Yeng Lin, Yu-Chen Wang, Long-Sheng Chen, Chia-Lun Kuo, Wen-Hsuan Hou.

**Project administration:** Chien-Yeh Lu, Yi-Chin Huang.

**Supervision:** Chung-Yi Li, Pei-Chen Lee, Ming-Yeng Lin, Yu-Chen Wang, Chia-Lun Kuo, Wen-Hsuan Hou.

**Writing – original draft:** I-Chen Chen, Chung-Yi Li, Wen-Hsuan Hou.

**Writing – review & editing:** I-Chen Chen, Chung-Yi Li, Chien-Yeh Lu, Yi-Chin Huang, Pei-Chen Lee, Ming-Yeng Lin, Yu-Chen Wang, Long-Sheng Chen, Chia-Lun Kuo, Wen-Hsuan Hou.

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
