## [Decision Letter · Decision Letter 0]

3 Feb 2023

PONE-D-22-16885Psychometric properties of novel instrument for evaluating ambient air pollution health literacy in adultsPLOS ONE

Dear Dr. Hou,

Thank you for submitting your manuscript to PLOS ONE. After careful consideration, we feel that it has merit but does not fully meet PLOS ONE’s publication criteria as it currently stands. Therefore, we invite you to submit a revised version of the manuscript that addresses the points raised during the review process.

We look forward to receiving your revised manuscript.

Kind regards,

Mohammad Asghari Jafarabadi

Academic Editor

PLOS ONE

Reviewers' comments:

Reviewer's Responses to Questions

**Comments to the Author**

1. Is the manuscript technically sound, and do the data support the conclusions?

Reviewer #1: Yes

2. Has the statistical analysis been performed appropriately and rigorously? 

Reviewer #1: Yes

3. Have the authors made all data underlying the findings in their manuscript fully available?

Reviewer #1: Yes

4. Is the manuscript presented in an intelligible fashion and written in standard English?

Reviewer #1: Yes

5. Review Comments to the Author

Reviewer #1: The authors developed a questionnaire on health literacy of ambient air pollution and validated the tool in a sample of > 1000 Taiwanese adults. They find good psychometric properties of their tool, and lowest health literary scores in the dimension “appraising in disease prevention”. They conclude that the questionnaire represents a useful tool in guiding public health interventions.

The underlying study aim is highly relevant, and the development and validation of the questionnaire seem to be valid. I have some concerns about the presentation of the data, though.

• The sample size is large enough to support subgroup analysis, both for validation of the questionnaire as well as final health literacy scores. At least sex-stratified and subgroup analyses for the older, more vulnerable subgroup should be presented.

• Could you expand a bit more on the potential redundancy of items. As far as I understood, the consensus that all items are necessary was based on the experts’ evaluation plus the value of the factor loadings (> 0.5) However, such a high Cronbach alpha (0.93) could point to potential redundancy. The structural evaluation showed that 12 dimensions fitted best, but not how many items were needed for each dimension. Please discuss further.

• Within dimensions, how was the sequence of items determined? Judging from the English translation in Table 3, e.g. items 1 and 2, items 3 and 4, and items 7 and 8 seem to go from specific to general, whereas items 11 and 12, and items 13 and 14 seem to go from general to specific.

• English language needs some polishing. “More than half of the participants were married female” – I’m not quite sure what this sentence is trying to say, or why it is a problem. This should be elaborated by pointing out the distribution of sex and marital status in the general population. Or L 323: “The AAPHL instrument can provide stakeholders with the profiles of capacities and the needs of the public.” What does that mean, exactly?

• Was the information if participants resided in rural vs urban environment available? If not, please add to limitations.

Other comments:

One digit after comma should suffice for Table 1, in particular for proportions.

L153: are these the same as in the first interview?

L173: Institutional Review Board of which institution?

L 315: please provide more context, e.g. elaborate how the difficulty in appraising looks like in the cited references

L 317: “the results”: does this pertain to results from previous studies? Please elaborate.

6. PLOS authors have the option to publish the peer review history of their article (what does this mean?). If published, this will include your full peer review and any attached files.

Reviewer #1: No

---

## [Author Response · Author response to Decision Letter 0]

23 Feb 2023

We thank the reviewer for his/her comments and feedback on our manuscript. We have revised the manuscript based on the comments and believe it has made the paper stronger.

Reviewer #1: The authors developed a questionnaire on health literacy of ambient air pollution and validated the tool in a sample of > 1000 Taiwanese adults. They find good psychometric properties of their tool, and lowest health literary scores in the dimension “appraising in disease prevention”. They conclude that the questionnaire represents a useful tool in guiding public health interventions.

Thank you for your comments.

The underlying study aim is highly relevant, and the development and validation of the questionnaire seem to be valid. I have some concerns about the presentation of the data, though.

• The sample size is large enough to support subgroup analysis, both for validation of the questionnaire as well as final health literacy scores. At least sex-stratified and subgroup analyses for the older, more vulnerable subgroup should be presented.

Thank you for your comments. We have conducted sex-stratified and age subgroup analyses for validation of the questionnaire and health literacy scores. The results were described in the result section and presented as supporting information following the manuscript.

“Sex-stratified and age subgroup analyses of model fits were presented in S1 and S2 Tables, respectively. Subgroups of male and female demonstrated similar trends in the three different models and the model fits of 12-factor model were acceptable. Subgroup of aged younger than 65 showed that model fits of 12-factor model was fair. On the other hand, subgroup of aged 65 or over did not demonstrate acceptable model fits in any models.”

“Sex-stratified and age subgroup analyses of scores were presented in S3 and S4 Tables, respectively. Subgroups of male and female demonstrated similar performance across the 12 dimensions and the dimension with highest score was applying in disease prevention. Subgroup of aged 65 or over showed higher total score than subgroup of aged younger than 65. The dimension with highest score in the subgroup of aged 65 or over was applying in health promotion while the subgroup of aged younger than 65 had the highest score in applying in disease prevention.”

• Could you expand a bit more on the potential redundancy of items. As far as I understood, the consensus that all items are necessary was based on the experts’ evaluation plus the value of the factor loadings (> 0.5) However, such a high Cronbach alpha (0.93) could point to potential redundancy. The structural evaluation showed that 12 dimensions fitted best, but not how many items were needed for each dimension. Please discuss further.

Thank you for your comments. We have added further discussion about the potential redundancy in the discussion section. 

“Bland and Altman suggest that critical values of Cronbach’s alpha may vary, which depends on different situations, and that value between 0.90 and 0.95 is desirable for clinical application [29]. However, Streiner suggest that values of Cronbach’s alpha greater than 0.90 may be too high and implies the potential redundancy among items [30].”

• Within dimensions, how was the sequence of items determined? Judging from the English translation in Table 3, e.g. items 1 and 2, items 3 and 4, and items 7 and 8 seem to go from specific to general, whereas items 11 and 12, and items 13 and 14 seem to go from general to specific.

Thank you for your comments. Even though the sequence of items was determined without specific concerns within a dimension, the dimensions and items were ordered from individual level to population level, which followed the concept of European Health Literacy Survey Questionnaire.

• English language needs some polishing. “More than half of the participants were married female” – I’m not quite sure what this sentence is trying to say, or why it is a problem. This should be elaborated by pointing out the distribution of sex and marital status in the general population. Or L 323: “The AAPHL instrument can provide stakeholders with the profiles of capacities and the needs of the public.” What does that mean, exactly?

Thank you for your comments. We have addressed the language concerns and elaborate more details in the discussion section.

“According to data of the Department of Household Registration in 2021, 50.47%, 43.76%, and 38.59% of the population are female, married, and had college or higher educational level, respectively [37]. However, 55.59%, 65.00%, and 61.54% of the participants were female, married, and had college or higher educational level, respectively, which might be due to selective response.”

“The AAPHL instrument can provide stakeholders with the profiles of capacities and the needs of the public, which provides information about what interventions might be needed to ameliorate health outcomes and new insight into how to improve AAPHL of the public.”

• Was the information if participants resided in rural vs urban environment available? If not, please add to limitations.

Thank you for your comments. We have addressed the unavailable information about participants’ residential environment as one of the study limitations in the discussion section. 

“Lastly, while EHL varies with cultural contexts and living environments, information regarding where participants resided in, either rural or urban environment, is unavailable.”

Other comments:

- One digit after comma should suffice for Table 1, in particular for proportions.

Thank you for your comments. We have revised the values with one digit after comma in Table 1.

- L153: are these the same as in the first interview?

They were different interviewees in the two cognitive interviews. We have added more details in the materials and methods section.

“Then, we invited 16 interviewees who were above 20 years of age and not involved in the first interview for the second face-to-face cognitive interviews.”

- L173: Institutional Review Board of which institution?

We have added more details in the materials and methods section. 

“The ethical approval of the final version of AAPHL instrument was obtained from Institutional Review Board of National Cheng Kung University Governance Framework for Human Research Ethics (#109-385).”

- L 315: please provide more context, e.g., elaborate how the difficulty in appraising looks like in the cited references

Thank you for your comments. We have added more descriptions about the difficulty in the cited references.

“Our finding is consistent with the previous studies in Japan and Europe, which show appraising health information is the most difficult competency. Appraising information is considered as a complex health literacy competency. Although the valuable and reliable websites exist, these are not always accessible, understandable, or usable by the public with low health literacy. Then, it may become difficult for the public to judge information regarding health [18,31,32].”

- L 317: “the results”: does this pertain to results from previous studies? Please elaborate.

Thank you for your comments. We have provided more details about the results from previous studies.

“Yet, our finding conflicts with the prior studies, which demonstrate the best competency is understanding health information. Information-processing of understanding is considered as a basic health literacy competency. Understanding health information is highly associated with educational level and socio-economic characteristics [18,31,32].”

---

## [Decision Letter · Decision Letter 1]

13 Apr 2023

Psychometric properties of novel instrument for evaluating ambient air pollution health literacy in adults

PONE-D-22-16885R1

Dear Dr. Hou,

We’re pleased to inform you that your manuscript has been judged scientifically suitable for publication and will be formally accepted for publication once it meets all outstanding technical requirements.

Kind regards,

Mohammad Asghari Jafarabadi

Academic Editor

PLOS ONE

Reviewers' comments:

Reviewer's Responses to Questions

**Comments to the Author**

1. If the authors have adequately addressed your comments raised in a previous round of review and you feel that this manuscript is now acceptable for publication, you may indicate that here to bypass the “Comments to the Author” section, enter your conflict of interest statement in the “Confidential to Editor” section, and submit your "Accept" recommendation.

Reviewer #1: All comments have been addressed

2. Is the manuscript technically sound, and do the data support the conclusions?

Reviewer #1: Yes

3. Has the statistical analysis been performed appropriately and rigorously? 

Reviewer #1: Yes

4. Have the authors made all data underlying the findings in their manuscript fully available?

Reviewer #1: Yes

5. Is the manuscript presented in an intelligible fashion and written in standard English?

Reviewer #1: Yes

6. Review Comments to the Author

Reviewer #1: The authors‘ revised manuscript is appreciated. My points were sufficiently addressed, no further comments.

7. PLOS authors have the option to publish the peer review history of their article (what does this mean?). If published, this will include your full peer review and any attached files.

Reviewer #1: No

---

## [Editor Report · Acceptance letter]

9 Jun 2023

PONE-D-22-16885R1 

Psychometric properties of novel instrument for evaluating ambient air pollution health literacy in adults 

Dear Dr. Hou:

I'm pleased to inform you that your manuscript has been deemed suitable for publication in PLOS ONE. Congratulations! Your manuscript is now with our production department. 

Kind regards, 

on behalf of

Professor Mohammad Asghari Jafarabadi 

Academic Editor

PLOS ONE